# Dopamine receptor activation regulates reward expectancy signals during cognitive control in primate prefrontal neurons

Torben Ott [1,2] ✉, Anna Marlina Stein[1] & Andreas Nieder [1] ✉

Dopamine neurons respond to reward-predicting cues but also modulate information processing in the prefrontal cortex essential for cognitive control. Whether dopamine controls reward expectation signals in prefrontal cortex that motivate cognitive control is unknown. We trained two male macaques on a working memory task while varying the reward size earned for successful task completion. We recorded neurons in lateral prefrontal cortex while simultaneously stimulating dopamine D1 receptor (D1R) or D2 receptor (D2R) families using micro-iontophoresis. We show that many neurons predict reward size throughout the trial. D1R stimulation showed mixed effects following reward cues but decreased reward expectancy coding during the memory delay. By contrast, D2R stimulation increased reward expectancy coding in multiple task periods, including cueing and memory periods. Stimulation of either dopamine receptors increased the neurons' selective responses to reward size upon reward delivery. The differential modulation of reward expectancy by dopamine receptors suggests that dopamine regulates reward expectancy necessary for successful cognitive control.

The primate dorsolateral prefrontal cortex (dlPFC) implements cognitive control via a range of cognitive subprocesses[1]. Neurons in dlPFC are engaged in high-level functions such as working memory, endogenous attention, rule switching, and planning[2–4]. Such cognitive processes are effortful and usually require the prospect of reward to be executed by subjects[5]. The expectation of reward acts as a motivational signal to overcome the effort, or cognitive costs, associated with cognitive control. Reward expectation must thus be maintained during task performance.

Indeed, high reward expectation can guide goal-directed behavior by improving cognitive control performance[6,7]. In line with behavioral observations, reward expectation modulates neural activity related to cognitive control subprocesses in dlPFC[8–11]. One such process is working memory, the maintenance and manipulation of information, which is encoded by delay activity of dlPFC neurons[12–16]. When monkeys performed a spatial working memory task in which a reward cue

at the beginning of the trial indicated how much juice the animal would receive for correct choices, neurons in frontal cortex and dlPFC encode both the expected reward as well as the spatial information needed for movement preparation[17–19]. Expected reward can modulate neuronal encoding of cognitive factors such as spatial memory information or object categories[18,20,21]. Based on these findings, it is suggested that dlPFC uses value information to guide working memory[22] and regulate costly cognitive control[23].

Working memory and other cognitive control processes in dlPFC are influenced by dopamine, a neuromodulator with a well-established role in reward signaling and learning[24]. In dlPFC, two main dopamine receptor families, D1R and D2R, mediate dopamine's control of neuronal activity. These dopamine receptor families play different, often complementary roles in gating sensory signals[25,26] working memory-related processes (both D1R[27–34] and D2R[32,35–37]), association learning[38–40], and the control of motor output[35,37,41]. Dopamine-releasing

[1]Animal Physiology, Institute of Neurobiology, Auf der Morgenstelle 28, University of Tübingen, 72076 Tübingen, Germany. [2]Present address: Bernstein Center for Computational Neuroscience and Institute of Biology, Humboldt-University of Berlin, 10099 Berlin, Germany. ✉e-mail: torben.ott@bccn-berlin.de; andreas.nieder@uni-tuebingen.de

neurons exhibit bursts of activity in response to reward-predicting cues or other salient events[24,42]. Therefore, dopamine has been hypothesized to motivate costly cognitive control in dlPFC[23]. However, it is unknown if and how dopamine receptors control the reward expectation signals in dlPFC that could integrate information about upcoming rewards with sustained cognitive control signals. Understanding dopamine's likely role in the interaction between reward signals and cognitive control signals would help to disentangle dopamine's diverse computational roles in motivating goal-directed behavior and their dysfunction in psychiatric diseases.

In the current study, we explored the role of dopamine when reward expectation and working memory demands are combined during a delayed match-to-sample task used to study working memory. We hypothesized that both D1Rs and D2Rs regulate reward expectancy coding of dlPFC neurons during working memory. To that aim, we investigated the activity of individual dlPFC neurons in rhesus monkeys by selectively activating D1Rs or D2Rs in the dlPFC.

## Results

Two male macaque monkeys were trained on a delayed match-to-sample task in which sample images had to be memorized and matched to the subsequently shown test images (Fig. 1A). A reward cue at the beginning of each trial signaled the amount of fluid reward the monkey would receive after successful completion of this working memory task. The two possible reward amounts (small or large) were cued by two different types of cues to dissociate reward expectancy signals from sensory signals related to cue appearance: a blue square or a grey X-shape indicated that the monkey could expect a small

reward, whereas a red square or a grey circle signaled that the monkey could anticipate a large reward (Fig. 1B).

## Behavior

To verify that the monkeys had learned the meaning of the reward-predicting cues, we analyzed the monkeys' behavior in recording sessions. We reasoned that the monkeys would be more motivated to correctly complete a trial that promised a large reward compared to trials resulting in a small reward. Indeed, performance accuracy (percentage of correct trials) was lower in trials with small reward expectancy in comparison to trials with large reward expectancy for both monkeys (Fig. 1C, D, Δ performance = 3.1% ± 0.3% (large minus small reward), $n = 79$ sessions, $p < 10^{-6}$, ANOVA, for monkey 1; Δ performance = 3.2% ± 0.4%, $n = 80$, $p < 10^{-5}$ for monkey 2) and was not influenced by reward cue set ($p > 0.2$ for main factor reward cue set or interactions between reward size and reward cue set). In addition, the percentage of aborted trials, in which monkeys broke eye fixation, was larger in trials with small reward expectancy compared to trials with large reward expectancy for both monkeys (Fig. 1E, F, Δ breaks = −17% ± 0.9%, $n = 79$, $p < 10^{-10}$, for monkey 1; Δ breaks = −14% ± 1.5%, $n = 80$, $p < 10^{-10}$, for monkey 2), and was also not influenced by reward cue set ($p > 0.2$ for all other comparisons). Finally, reaction times (RTs) of both monkeys were longer in trials with small reward expectancy compared to trials with large reward expectancy (Fig. 1G, H, Δ RT = −42 ms ± 2 ms, $n = 79$, $p < 10^{-10}$, for monkey 1; Δ RT = −38 ms ± 5 ms, $n = 80$, $p < 10^{-10}$, for monkey 2), again independent of reward cue set ($p > 0.7$ for all other comparisons). These three behavioral parameters (performance accuracy, aborted trials, and RT)

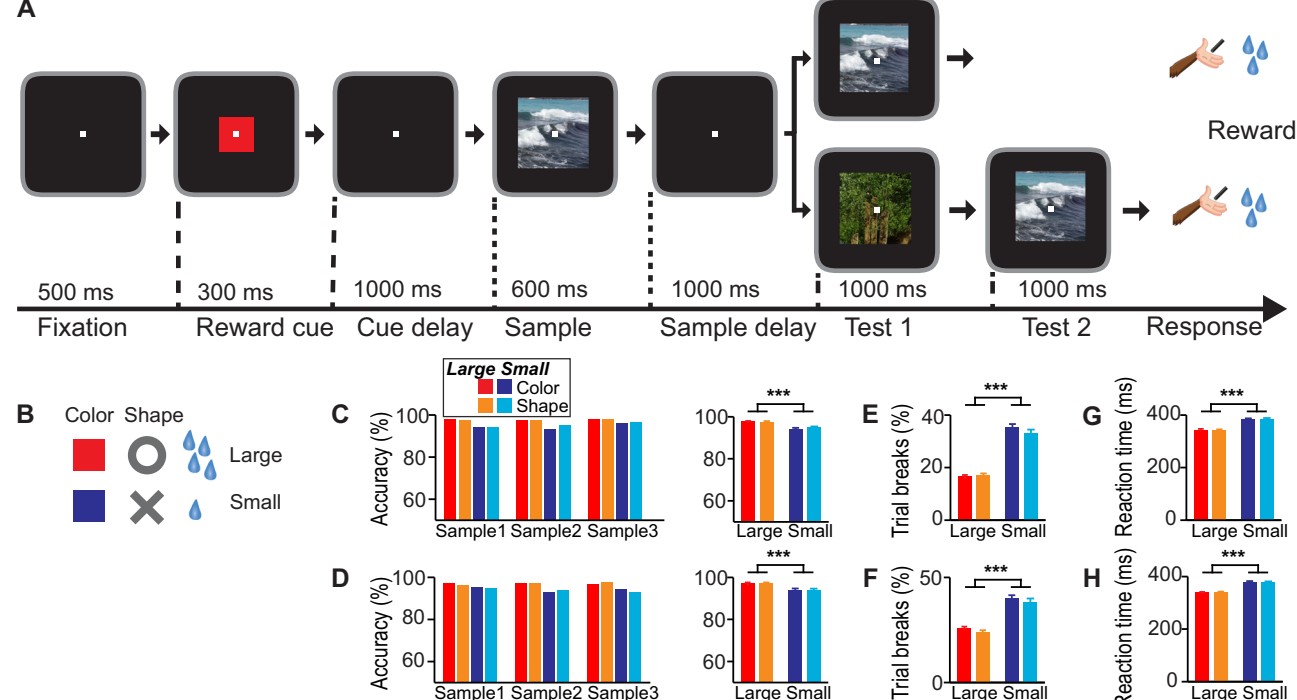

**Fig. 1 | Monkeys acquired expectations about cued reward size in a working memory task. A** Monkeys initiated a trial by grabbing a lever and fixating a central fixation spot, which had to be maintained throughout the trial. After a pure fixation period, a reward cue predicted the amount of liquid reward the monkeys received at the end of a trial for a correct choice (small or large). After a cue delay period, a visual sample stimulus appeared on the screen, which monkeys had to memorize during the sample delay period. In the test period, monkeys had to release the lever if the same stimulus appeared (50% of trials) and to keep holding the lever if a different stimulus appeared (50% of trials) to receive the cued liquid reward amount. **B** Two sets of cues indicated the reward size, a color set (red square for

large reward, blue square for small reward) and a shape set (gray annulus for large reward, gray cross for small reward). **C** Behavioral performance for monkey 1 for all different conditions (left, single session example). Performance was lower on small reward trials (right, all recording sessions) ($p < 10^{-6}$, ANOVA, $n = 79$ sessions). **D** Same conventions as in (**C**) for monkey 2 ($p < 10^{-5}$, ANOVA, $n = 80$). **E** Percentage of aborted trials (i.e., trials in which the monkey broke eye fixation) for monkey 1 was higher for small reward trials ($p < 10^{-10}$). **F** Same conventions as in (**E**) for monkey 2 ($p < 10^{-10}$). **G** Reaction times of monkey 1 were higher for small reward trials ($p < 10^{-10}$). **H** Same conventions as in **G** for monkey 2 ($p < 10^{-10}$). *** $p < 0.001$ (ANOVA).

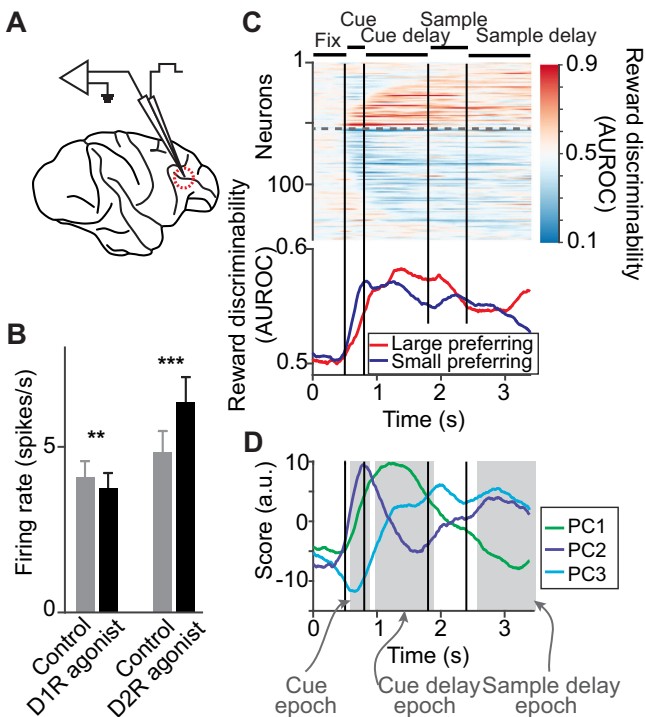

**Fig. 2 | Single neurons in dlPFC signal reward expectation across multiple task periods. A** Recording site in dlPFC (area 46) of two macaque monkeys. **B** Effects of D1R stimulation with SKF81297 (129 neurons, $p = 0.008$) and D2R stimulation with quinpirole (127 neurons; $p < 10^{-6}$) on neuronal firing rate in the baseline period. Data are presented as mean values +/- SEM. (signed-rank test, two-sided, ***$p < 0.001$, **$p < 0.01$) **C** Time-resolved reward discriminability quantified via AUROC between spike rate distributions in large and small reward trials throughout the task. Neurons are sorted by the latency of reward discrimination (see Methods) and grouped by their overall reward size preference (above dashed line, overall large-preferring; determined by the average firing rate between reward cue onset and test stimulus onset). **D** First three temporal components explaining the largest temporal variability in temporal reward discrimination in all recorded neurons. For subsequent analyses, we defined 3 main task periods (gray shaded areas, see text and Methods for details).

indicated that the monkeys understood the meaning of the reward cues and that their reward expectation modulated working performance.

## Single-neuron recordings combined with micro-iontophoretic drug application

We recorded 256 single units in 159 recording sessions (79 for monkey 1, 80 for monkey 2) from the dlPFC centered around the principal sulcus of both monkeys (Fig. 2A), while the monkeys were performing the task. Here, we focused on dopamine receptor modulation of reward expectancy signals – dopamine receptor modulation of working memory signals in similar behavioral tasks were reported previously[32,33,36]. To directly assess the impact of dopamine receptor targeting agents on neuronal reward expectancy signals, each neuron was recorded both without drug application (*control condition*) and while applying dopamine receptor agents at the vicinity of the recorded neurons using micro-iontophoresis (*drug condition*). Control conditions alternated with drug conditions in each recording session. In each session, we tested one of two different substances that selectively targeted the D1R or the D2R: The D1R was assessed in 129 neurons by applying the D1R agonist SKF81297. The D2R was tested in 127 neurons using the D2R agonist quinpirole. In previous experiments, we could exclude any effect on neuronal firing properties when applying normal saline with comparable injection currents[25,32]. As in previous

experiments[32,33], we found that D1R stimulation slightly decreased the baseline firing rates of dlPFC neurons (Δ firing rate = −0.33 ± 0.13 spikes/s, $p = 0.008$, signed-rank test), whereas D2R stimulation increased the baseline firing rates of dlPFC neurons (Δ firing rate = +1.50 ± 0.29 spikes/s, $p < 10^{-6}$) (Fig. 2B).

### Reward expectancy neurons

We identified neurons with selective responses for reward size before reward delivery ("reward expectancy neurons") in three non-overlapping analysis windows: (i) the cue period (400 ms, beginning 100 ms after reward cue onset to account for average neuronal response latencies), (ii) the cue delay period (900 ms, beginning 200 ms after reward cue offset), and (iii) the sample delay period (900 ms, beginning 200 ms after memory sample offset) (Fig. 2C, see shaded trial periods in Fig. 2D). To that aim, we used analysis of variance (ANOVA) with main factors 'reward size' (large/small), 'reward cue set' (color/shape), iontophoretic 'drug application' (control condition/drug condition), and 'memorized sample' (sample delay epoch only). Neurons with a significant main effect of reward size ($p < 0.05$) were included in subsequent analyses. To analyze dopamine receptor modulation of reward expectancy signals irrespective of the visual properties of the reward cues, we excluded neurons with significant main effect for reward cue set or an interaction between reward size and reward cue set ($p < 0.05$). This analysis revealed that about half of the neurons encoded reward expectancy in cue and/or delay periods (145/256, not corrected for multiple comparisons) (Table 1). More neurons preferred the small reward size, i.e., had a larger firing rate during trials with small reward expectancy as compared to trials with large reward expectancy, in the cue period (6% vs. 16 %, $p = 0.0003$, Chi-square-test) and cue delay period (11% vs. 18 %, $p = 0.04$), but not in the sample delay period (13% vs. 18%, $p = 0.2$) (Table 1).

We first quantified the coding quality of reward expectancy throughout the trial using a time-resolved reward discriminability index (AUROC, the area under the receiver operator characteristic derived from signal detection theory). Values of 0.5 correspond to an absence of coding. Values larger than 0.5 indicated stronger coding quality for large reward-preferring neurons, whereas values smaller than 0.5 indicated stronger coding quality for small reward-preferring neurons. Reward expectancy neurons encoded reward size at different task periods, with different neurons showing transient or sustained reward coding throughout the trial (Fig. 2C). To characterize typical time courses for reward size coding in our population data, we visualized the three first principal components that explained most of the temporal variability observed in time-resolved reward coding including all recorded neurons irrespective of coding properties (Fig. 2D). Most neurons showed preferential coding for one of the temporal components (sparse coding, see Methods), suggesting that these components reflect typical reward coding dynamics. Prominent reward coding dynamics were (i) phasic reward coding after reward cue onset, (ii) semi-sustained reward coding in the cue delay period, and (iii) sustained reward coding throughout the sample delay period. These temporal patterns helped to define the three main analysis windows to test whether dopamine receptor modulated neuronal reward expectancy signals.

### Reward cue period: D2R stimulation, but not D1R stimulation, enhances reward expectancy signals

We compared the quality of coding of reward expectancy neurons in the control (no-drug) condition with the D1R stimulation (SKF81297 application) condition during the reward cue period. A representative reward expectancy neuron was barely influenced by D1R stimulation (Fig. 3A). We constructed population responses by pooling trials within each reward cue set (i.e., small versus large) and averaging normalized activity to the preferred and nonpreferred reward size, defined as the reward size yielding the larger and lower firing rate, respectively. The

**Table 1 | Number of reward expectancy neurons**

|  |  | Cue | Cue delay | Sample delay | Nonmatch | Reward | All |
|---|---|---|---|---|---|---|---|
| SKF81297 sessions | (large, small) | 30 (9, 21) | 31 (15, 16) | 33 (12, 21) | 25 (8, 17) | 21 (14, 7) | 129 |
| Quinpirole sessions | (large, small) | 28 (7, 21) | 43 (14, 29) | 46 (22, 24) | 27 (9, 18) | 20 (7, 13) | 127 |
| ∑ |  | 58 (23 %) | 74 (29%) | 79 (31%) | 52 (20%) | 41 (16%) | 256 |

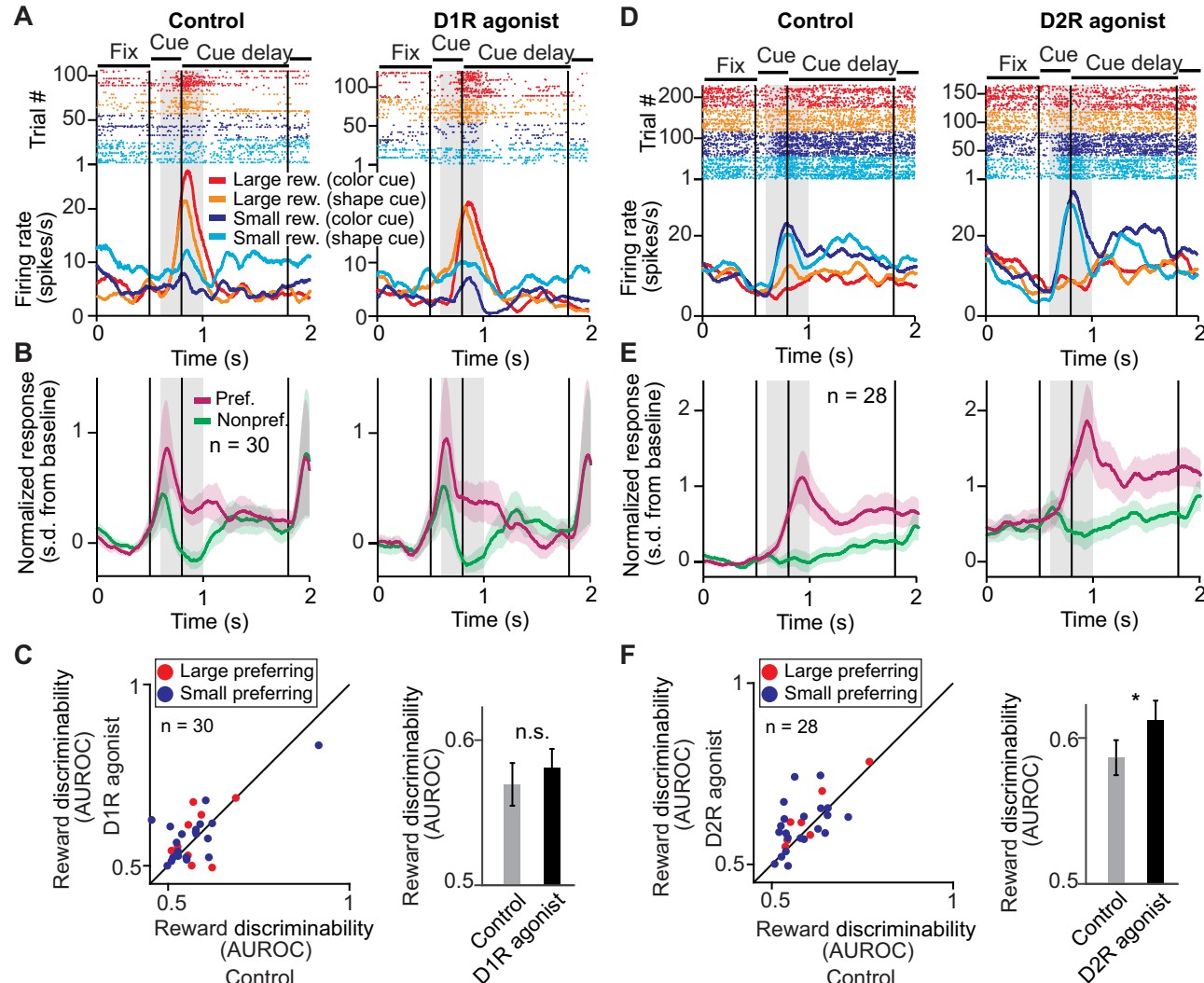

**Fig. 3 | D2R acitivation, but not D1R activation, increased neuronal reward expectancy signals during the cue period. A** Example reward expectancy neuron recorded during control conditions (left) and after stimulating D1Rs with SKF81297 (right) with higher activity for cues indicating large reward in the cue period (gray shaded area, cue period). **B** Average normalized activity of all reward expectancy neurons in the cue period (gray shaded area) recorded with SKF81297. Activity was pooled over reward cue sets, and the preferred reward size was defined as the reward size condition yielding higher average activity. Error bands represent +/- SEM. **C** Reward discriminability (AUROC) in the cue period compared between control and D1R stimulation. Left: Each point corresponds to one neuron. Right: Average reward discriminability (AUROC) for control and D1R stimulation (30 neurons). Data are presented as mean values +/- SEM. **D** Example reward expectancy neuron recorded during control conditions (left) and after stimulating D2Rs with quinpirole (right) with higher activity for cues indicating small reward. **E** Same conventions as in (**B**) for all reward expectancy neurons recorded with quinpirole. Error bands represent +/- SEM. **F** Reward discriminability (AUROC) in the cue period compared between control and D2R stimulation. Left: Each point corresponds to one neuron. Right: Average reward discriminability (AUROC) for control and D2R stimulation (28 neurons, $p = 0.05$). Data are presented as mean values +/- SEM. $^*p \le 0.05$, n.s. not significant ($p > 0.05$), signed rank test.

firing rates of the population of all reward-selective neurons recorded during sessions of D1R stimulation with SKF81297 were unchanged (Fig. 3B). Consistent with these firing rate observations, D1R stimulation with SKF81297 did not systematically change reward discriminability in the cue period (Fig. 3C, Δ AUROC = +0.01 ± 0.01, $n = 30$, $p = 0.21$, signed rank test, two-sided).

In contrast, D2R stimulation with quinpirole of another reward expectancy neuron selective for small reward showed a prominent increase in reward selectivity (Fig. 3D). This effect was systematically observed for the population activity of all neurons recorded during sessions with quinpirole application (Fig. 3E). Consequently, D2R stimulation with quinpirole significantly increased reward discriminability in the cue period (Fig. 3F, Δ AUROC = +0.03 ± 0.01, $n = 28$, $p = 0.05$, signed rank test, two-sided). Thus, D2R stimulation, but not D1R stimulation, increased reward expectancy coding of single neurons during the cue period, when the cues were presented.

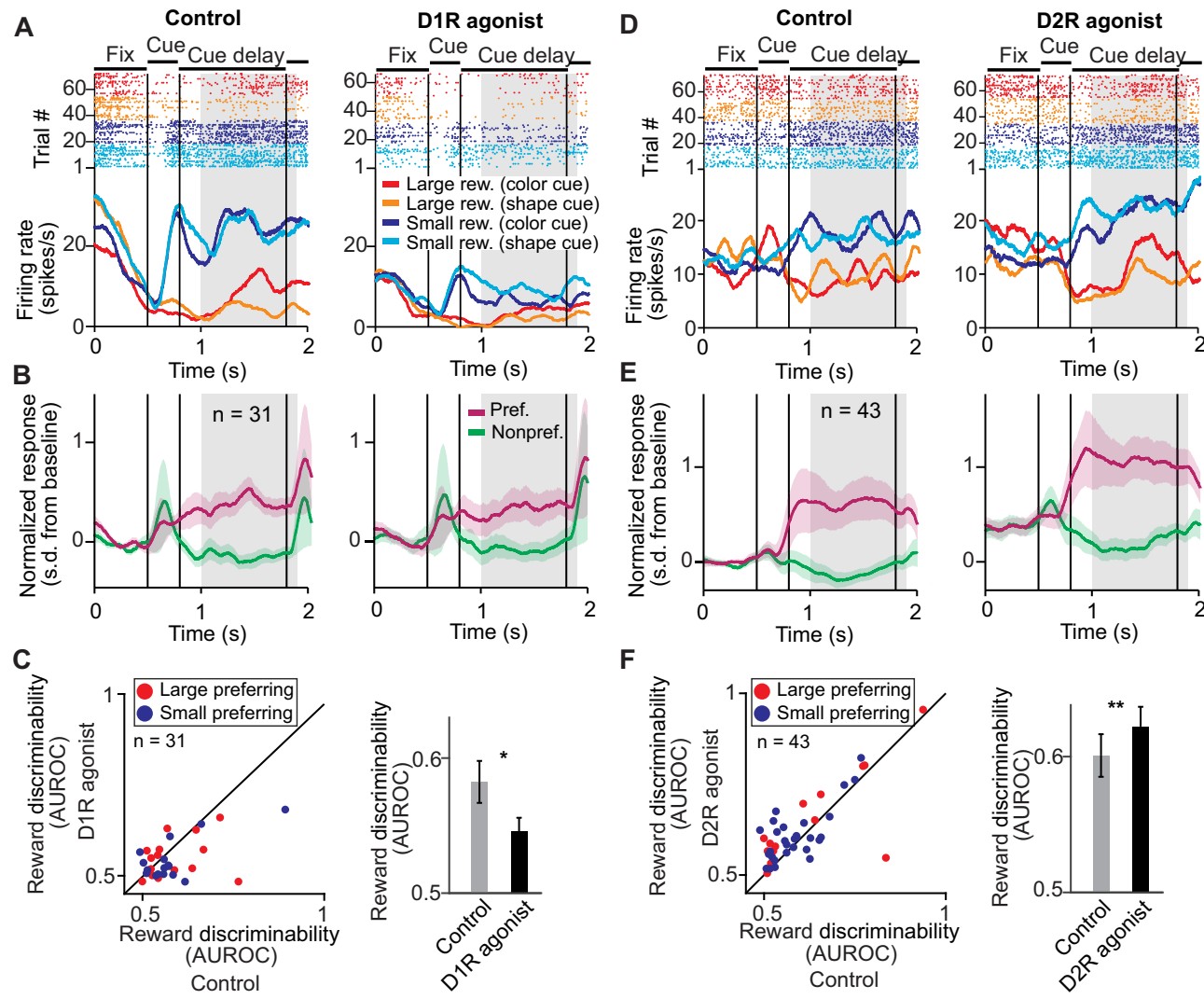

**Fig. 4 | D1R and D2R oppositely modulated neuronal reward expectancy signals during the cue delay period. A** Example reward expectancy neuron recorded during control conditions (left) and after stimulating D1Rs with SKF81297 (right) with higher activity for cues indicating large reward in the cue delay period (gray shaded area, cue delay period). **B** Average normalized activity of all reward expectancy neurons in the cue delay period (gray shaded area) recorded with SKF81297. Activity was pooled over reward cue sets, and the preferred reward size was defined as the reward size condition yielding higher average activity. Error bands represent +/- SEM. **C** Reward discriminability (AUROC) in the cue delay period compared between control and D1R stimulation. Left: Each point corresponds to one neuron. Right:

Average reward discriminability (AUROC) for control and D1R stimulation (31 neurons; $p = 0.012$). Data are presented as mean values +/- SEM. **D** Example reward expectancy neuron recorded during control conditions (left) and after stimulating D2Rs with quinpirole (right) with higher activity for cues indicating small reward. **E** Same conventions as in **B** for all reward expectancy neurons recorded with quinpirole. Error bands represent +/- SEM. **F** Reward discriminability (AUROC) in the cue delay period compared between control and D2R stimulation. Left: Each point corresponds to one neuron. Right: Average reward discriminability (AUROC) for control and D2R stimulation (43 neurons; $p = 0.0016$). Data are presented as mean values +/- SEM. *$p < 0.01$, *$p < 0.05$, signed rank test.

## Cue delay period: D1R and D2R stimulation have opposite effects on reward expectancy signals

We repeated the same analyses during the cue delay period. Here, stimulating D1Rs with SKF81297 impaired the selectivity of an example reward expectancy neuron selective for small reward size (Fig. 4A). This effect was consistently observed for all selective neurons recorded in sessions with SKF81297 application, showing that D1R stimulation decreased selectivity for the neurons' preferred reward size (Fig. 4B). Accordingly, reward discriminability was impaired by D1R stimulation with SKF81297 (Fig. 4C, $\Delta$ AUROC $= -0.04 \pm 0.01$, $n = 31$, $p = 0.012$, signed rank test, two-sided), thus decreasing reward expectancy coding.

In contrast, D2R stimulation with quinpirole increased the selectivity for small rewards during the cue delay period of another reward expectancy neuron (Fig. 4D), as also observed for the population of all selective neurons recorded in sessions with quinpirole application

(Fig. 4E). In accordance, D2R stimulation significantly increased reward discriminability (Fig. 4F, $\Delta$AUROC $= +0.02 \pm 0.01$, $n = 43$, $p = 0.0016$, signed rank test, two-sided), thus enhancing the neurons' coding capacity for reward expectancy. In sum, D1Rs and D2Rs acted oppositely on reward expectancy coding of single neurons during the cue delay period.

## Sample delay period: D2R stimulation increases reward expectancy signals

We repeated the same analyses during the sample delay period. When the monkeys memorized the sample in conjunction with the expected reward size, we observed no clear modulation of reward expectancy signals following D1R stimulation with SKF81297. This can be seen in an example reward expectancy neuron that did not show altered reward size selectivity (Fig. 5A). The absence of an effect was confirmed for the population of reward expectancy neurons recorded in sessions with

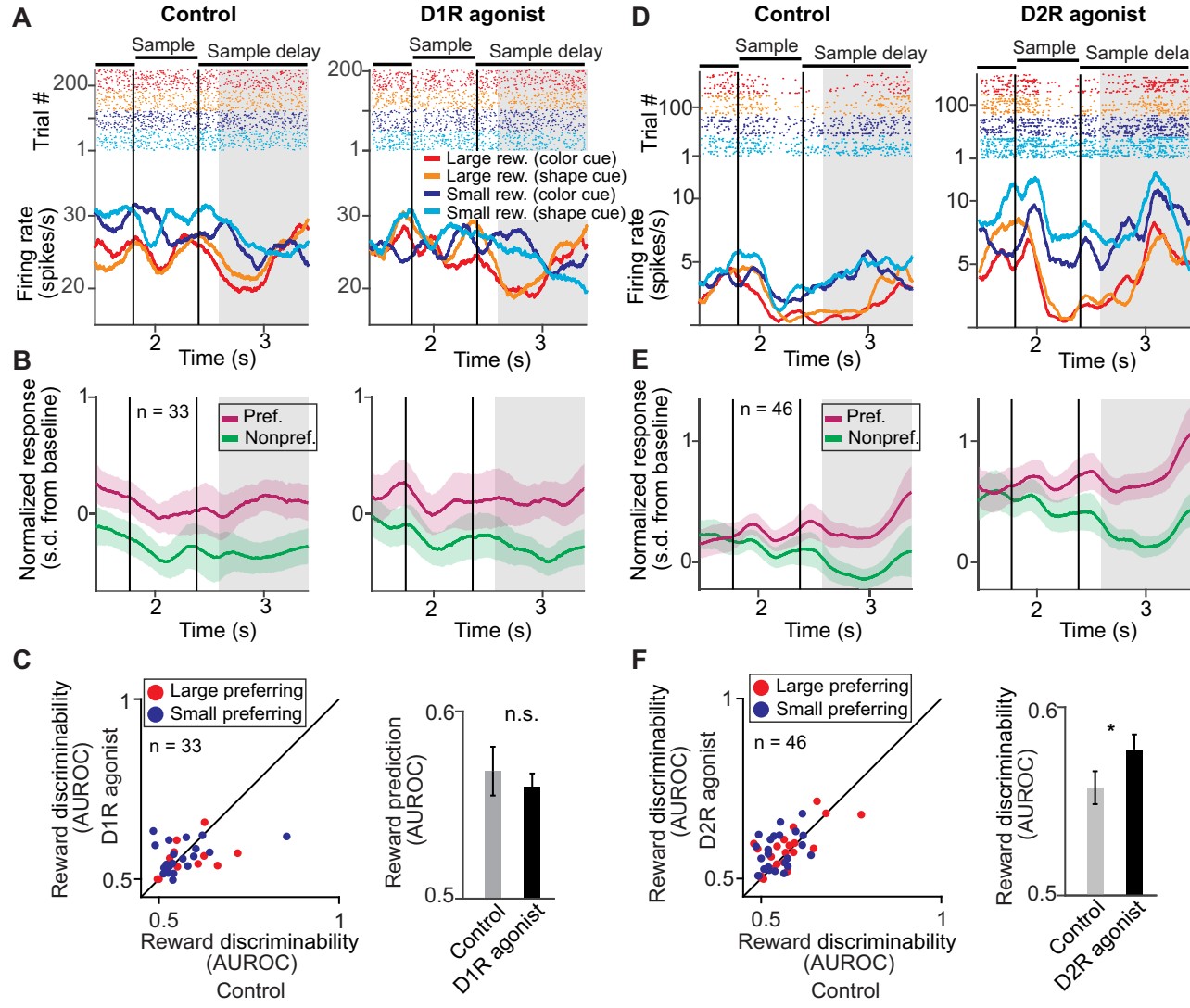

**Fig. 5 | D2R acitvation, but not D1R activation, increased neuronal reward expectancy signals during the sample delay period. A** Example reward expectancy neuron recorded during control conditions (left) and after stimulating D1Rs with SKF81297 (right) with higher activity for cues indicating large reward in the sample delay period (gray shaded area, sample delay period). **B** Average normalized activity of all reward expectancy neurons in the cue delay period (gray shaded area) recorded with SKF81297. Activity was pooled over reward cue sets, and the preferred reward size was defined as the reward size condition yielding higher average activity. Error bands represent +/- SEM. **C** Reward discriminability (AUROC) in the sample delay period compared between control and D1R stimulation. Left: Each point corresponds to one neuron. Right: Average reward discriminability

(AUROC) for control and D1R stimulation (33 neurons). Data are presented as mean values +/- SEM. **D** Example reward expectancy neuron recorded during control conditions (left) and after stimulating D2Rs with quinpirole (right) with higher activity for cues indicating small reward. **E** Same conventions as in (**B**) for all reward expectancy neurons recorded with quinpirole. Error bands represent +/- SEM. **F** Reward discriminability (AUROC) in the sample delay period compared between control and D2R stimulation. Left: Each point corresponds to one neuron. Right: Average reward discriminability (AUROC) for control and D2R stimulation (46 neurons). Data are presented as mean values +/- SEM. $^*p < 0.05$, n.s. not significant ($p > 0.05$), signed rank test.

SKF81297 application (Fig. 5B). Consistent with these observations, no significant changes in reward discriminability after D1R stimulation were detected (Fig. 5C, $\Delta$ AUROC $= -0.008 \pm 0.01$, $n = 33$, $p = 0.69$, signed rank test, two-sided).

However, similar to the cue delay period, D2R stimulation with quinpirole increased reward expectancy neurons' response to its preferred reward size, as can be seen for a small-reward preferring example neuron (Fig. 5D). This effect was mirrored in the population of reward expectancy neurons recorded in quinpirole sessions (Fig. 5E). Exploration of these neurons' reward discriminability confirmed that D2R stimulation on average increased reward discriminability (Fig. 5F, $\Delta$ AUROC $= +0.01 \pm 0.008$, $n = 46$, $p = 0.018$, signed rank test). In sum, D2R stimulation, but not D1R stimulation, increased neuronal reward expectancy signals in the sample delay

epoch, i.e., when the monkeys memorized the sample in conjunction with the expected reward size.

## Dopamine receptors differentially modulate population decoding of reward size

We next asked, based on decoding analyses, whether dopamine receptor stimulation modulated the population's capacity to represent the expected reward size. In the same three analysis windows as used before, we trained a linear decoder to predict the binary class "reward size" (small or large) by constructing population activity vectors of pseudo-simultaneous single trials using cross-validated test and training sets (see Methods). Overall, these decoders showed high performance with little misclassifications of reward size (decoder performance 80–90% across all analysis windows in control phases).

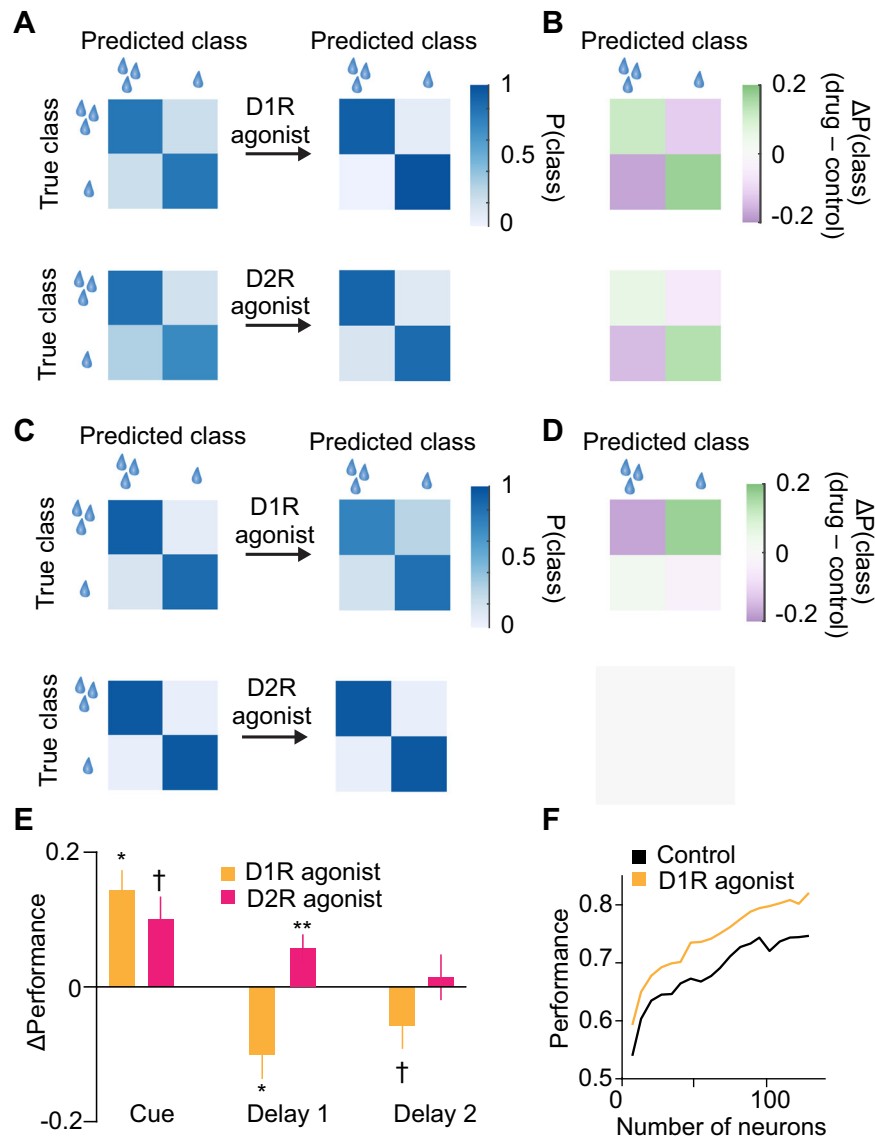

**Fig. 6 | D1R and D2R differentially modulated population decoding of reward size across task period. A** Performance (probability of the decoder predicting the correct reward size condition) of a linear decoder predicting the reward size class (large or small) of held-out test trials before (left) and after (right) D1R stimulation based on population activity in the cue period of all recorded neurons in SKF81297 sessions (top row) and quinpirole sessions (bottom row). **B** Decoder performance difference between control and drug conditions (top, D1R stimulation; bottom, D2R stimulation). **C** Same display as in (**A**) for the decoder's

performance based on population activity in the cue delay phase. **D** Same display as in (**B**) for the cue delay phase. **E** Summary and statistical evaluation of D1R (reward cue period: $n = 129$ neurons, $p = 0.014$; reward delay1 period: $p = 0.013$; sample delay2 period: $p < 0.1$) and D2R stimulation (reward cue period: $n = 127$ neurons, $p = 0.075$; reward delay1 period: $p = 0.004$; sample delay2 period: $p > 0.1$) of decoder performance across task periods. Data are presented as mean values +/- SEM. **F** Decoder performance in the cue period dependent on the number of sub-sampled neurons. $^{**}p < 0.01$, $^{*}p < 0.05$, $^{†}p < 0.1$, signed rank test.

During the reward cue period, both D1R and D2R stimulation increased the decoder's performance to predict reward size from population activity, showing an increase in correct and decrease in incorrect classifications across both classes. The absolute classification accuracies are shown in Fig. 6A, whereas classification accuracy relative to the no-drug control condition is depicted in Fig. 6B. D1R stimulation with SKF81297 significantly increased the decoder's overall performance (Fig. 6E, Δ performance = +0.14 ± 0.03 with $n = 129$ neurons, $p = 0.014$, shuffle test, two-sided). D2R stimulation displayed a trend for increased performance (Fig. 6E, Δ performance = +0.10 ± 7 with $n = 127$ neurons, $p = 0.075$).

In the reward delay period, D1R and D2R stimulation produced opposite effects on decoding performance (Fig. 6C, D): D1R stimulation decreased the decoder's performance (Fig. 6E, Δ performance = −0.10 ± 0.04 with $n = 129$ neurons, $p = 0.013$), while D2R

displayed a weak but statistically significant increase in decoder performance (Fig. 6E, Δ performance = +0.06 ± 0.02 with $n = 127$ neurons, $p = 0.004$). In the sample delay period, no clear change in decoding was observed after either D1R or D2R stimulation (Fig. 6E).

Since the observation that D1R stimulation increased decoding performance of reward size in the reward cue period is inconsistent with the lack of an effect on reward discriminability (cf. Fig. 3C), we asked whether the decoder's performance depends on the types or numbers of neurons included when training the decoder. Decoding performance after D1R stimulation increased across a large range of sampled neuronal sub-populations (Fig. 6F), suggesting that observed effects are robust and not dependent on a few neurons. Further, when using reward expectancy neurons only, no increase in the decoder's performance was observed (Δ performance = +0.01 ± 0.02 with $n = 30$ neurons, $p = 0.21$), while there was a statistical trend increase in the

decoder's performance when excluding reward expectancy neurons (Δ performance = +0.10 ± 0.04 with $n = 99$ neurons, $p = 0.059$). These results suggest that the decoder's performance increase after D1R stimulation was primarily driven by neurons not included in the reward discriminability analysis. D2R stimulation did not significantly increase the decoder's performance in the reward cue period after excluding reward expectancy neurons (Δ performance = +0.13 ± 0.04 with $n = 99$ neurons, $p = 0.13$). During the cue delay period, both D1R and D2R stimulation decreased the decoder's performance after excluding reward expectancy neurons (D1R: Δ performance = −0.16 ± 0.04 with $n = 98$ neurons, $p = 0.0030$; D2R: Δ performance = −0.16 ± 0.04 with $n = 84$ neurons, $p = 0.042$), a result consistent for D1R stimulation with the overall population, but inconsistent for D2R stimulation. This suggests that D2R's increase in reward size decoding was primarily driven by reward expectancy neurons. In the sample delay period, neither D1R nor D2R stimulation altered the decoder's performance ($p > 0.1$ each).

### Reward delivery: both dopamine receptors increase selective responses to reward size

Beyond exploring reward expectancy modulation, we asked whether dopamine receptors would also modulate neuronal responses to reward size after reward delivery for correct choices. We identified neurons selective to reward size with a 2-way ANOVA with main factors reward size (large/small) and iontophoresis condition (control/drug) that showed a significant main effect of reward size in a 500 ms window starting 100 ms after reward delivery (reward period) in correct match trials. Reward delivery followed a monkey's lever release as a response during the matching test stimulus (cf. Fig. 1A). Since both small and large reward required the same motor action with only the reward size differing, we reasoned that response differences indicated selectivity to delivered reward size. We identified 41 (16%) reward size selective neurons. A similar number of neurons responded with higher activity after large or small reward sizes (21 and 20 neurons, respectively, or 8% each, $p = 0.9$, Chi-square-test).

An example neuron with higher activity for large reward compared to small reward showed an increase in reward responses after D1R stimulation (Fig. 7A), even though, in general, D1R stimulation inhibited neuronal firing (cf. Fig. 2B). The population of all reward size selective neurons recorded in SKF81297 sessions displayed a moderate increase in responses to preferred reward size (Fig. 7B). Consistent with these observations, SKF81297 increased reward size responses quantified via the AUROC between spike rate distributions in large and small reward trials in the reward period (Fig. 7C, Δ AUROC = +0.03 ± 0.01, $n = 21$, $p = 0.04$, signed rank test, two-sided).

After D2R stimulation with quinpirole, neurons also showed elevated responses to reward as can be seen in an example neuron (Fig. 7D) but also in the population of neurons recorded in quinpirole sessions (Fig. 7E). D2R stimulation increased reward size responses, as well (Fig. 7F, ΔAUROC = +0.06 ± 0.02, $n = 20$, $p = 0.003$, signed rank test, two-sided). In sum, both dopamine receptors increased selective responses for reward size during reward delivery.

Finally, we asked how dopamine receptors modulate reward expectancy signals during the presentation of a non-matching test stimulus. In this task phase, the monkeys committed to a decision and are thus not required to remember the visual stimuli while anticipating reward delivery after lever release during subsequent presentation of the matching test stimulus, allowing us to analyze reward expectation signals in the absence of working memory load. A similar number of neurons were selective for reward size during the non-matching test period and, as in previous task epochs, more neurons preferred the small reward size (Table 1, 7% small preferring vs. 14% large-preferring, $p = 0.0085$, Chi-square-test). We observed no significant changes in reward discriminability after D1R stimulation during the non-matching test stimulus presentation (Fig. 7G, Δ AUROC = +0.0003 ± 0.02, $n = 25$,

$p = 0.81$, signed rank test, two-sided). D2R stimulation significantly increased reward discriminability (Fig. 7H, ΔAUROC = +0.036 ± 0.016, $n = 27$, $p = 0.046$, signed rank test, two-sided), thus enhancing the neurons' coding capacity for reward expectancy. Dopamine receptors thus showed a similar modulation of reward expectancy signals in the non-matching test stimulus than in previous task epochs.

## Discussion

We report that activating D1Rs and D2Rs modulates reward signals in dlPFC neurons in distinct ways. D2R stimulation increased reward expectancy signals throughout the trial and increased selective neuronal responses to reward size after reward delivery. On the other hand, D1R stimulation showed mixed effects, decreasing reward expectancy signals in the cue delay period, while increasing selective neuronal responses to reward size after reward delivery. These data suggest tailored dopamine receptor modulation of prefrontal circuits representing expected reward relevant for cognitive control.

Single neurons in the dlPFC represented expected reward magnitudes following presentation of the reward cue and during the delay period before and after sample presentation ("reward expectancy neurons"). Information about upcoming rewards is particularly relevant during cognitively demanding tasks, and previous studies have found that dlPFC neurons are more sensitive to value information during working memory performance[10,43,44]. Accordingly, representations of reward expectation in frontal cortex were found in a number of previous studies[8,9,18,43,45], and reward expectation modulated prefrontal visual[46] and spatial memory signals[8–11,18,47,48]. Given that dopamine neurons fire phasic bursts in response to salient sensory events[49,50] and, in particular, graded responses in response to cues predicting expected reward[51], it seems likely that the source of cortical reward-related signals stem from lateral midbrain dopamine neurons, which preferentially project to dlPFC[52]. However, since lateral midbrain dopamine neurons preferentially signal saliency instead of canonical reward prediction errors[42,44,53], the precise origin of dlPFC reward signals remains unclear.

Interestingly, more neurons were selective for small reward expectancies compared to large reward expectancies throughout reward anticipation, including the test stimulus period leading up to reward delivery, where cognitive load was reduced since no working memory was required. Given the continuous coding of reward expectancy as well as similar dopamine receptor stimulation throughout task epochs, the reward expectancy signals described here likely reflect a coding of relative value between large and small reward, rather than motivational factors alone[54]. Reward expectancy neurons described here were typically not selective for other features of the task such as the sample stimulus. In a previous study investigating spatial working memory and reward expectation[8], more neurons preferred the small reward size in the pool of neurons that did not show spatial selectivity, akin to our findings. Further, a previous study reported higher prefrontal dopamine levels following small reward as compared to large reward predictions[55]. We speculate that dopamine enables cognitive control by driving the motivation to obtain future rewards to overcome the cognitive costs associated with effortful cognition[23]. Small, but necessary, future rewards might require stronger activation of prefrontal dopamine to nevertheless mediate successful cognition or cope with stress[55].

As reported previously, D1R activation slightly suppressed neuronal firing in general[30,32]. Activating D1Rs did not systematically modulate reward discrimination in the cue period, contrary to previous findings that D1R stimulation increased the signal-to-noise ratio of sensory signals entering working memory[26,32]. However, decoding performance of reward size in the cue period increased after D1R stimulation, likely driven by the population of non- or weakly reward-size selective neurons. Thus, D1Rs might gate reward information reaching dlPFC populations, similar to sensory signals. Given these mixed

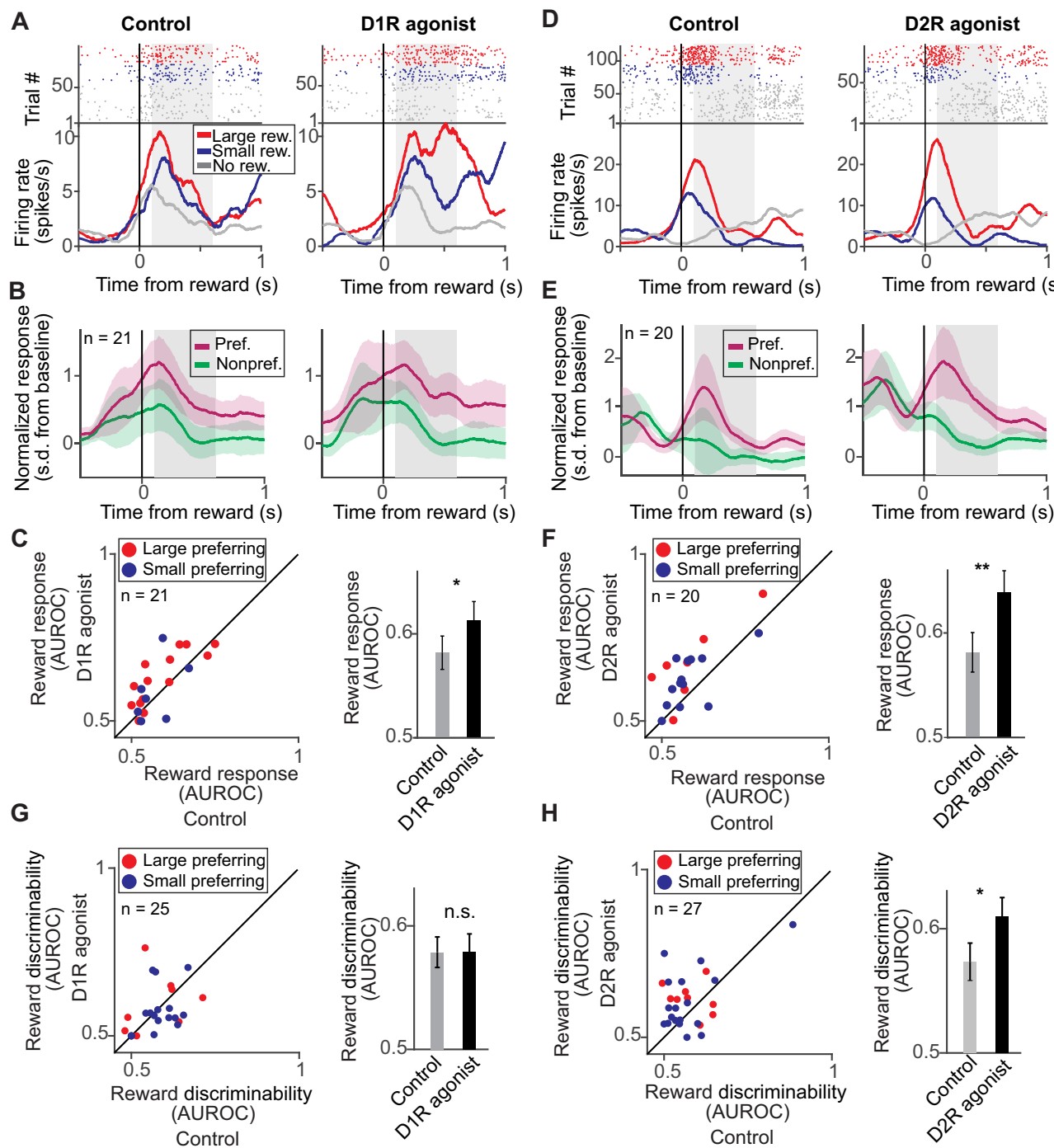

**Fig. 7 | D1R and D2R activation increased neuronal reward size selectivity after reward delivery and showed mixed effects in non-match periods. A** Example neuron selective for reward size after reward delivery recorded during control conditions (left) and after stimulating D1Rs with SKF81297 (right) with higher activity after large rewards (gray shaded area, reward period). **B** Average normalized activity of neurons selective for reward size in the reward delivery period (gray shaded area) recorded with SKF81297. Activity was pooled over reward cue sets, and the preferred reward size was defined as the reward size condition yielding higher average activity. Error bands represent +/- SEM. **C** Reward size response (AUROC) after reward delivery compared between control and D1R stimulation. Left: Each point corresponds to one neuron. Right: Average reward discriminability (AUROC) for control and D1R stimulation (21 neurons, *p* = 0.04). Data are presented as mean values +/- SEM. **D** Example reward neuron selective for reward size after reward delivery recorded during control conditions (left) and after stimulating D2Rs with quinpirole (right) with higher activity after large rewards. **E** Same

conventions as in (**B**) for all neurons selective for reward size in the reward period recorded with quinpirole. Error bands represent +/- SEM. **F** Reward size response (AUROC) after reward delivery compared between control and D2R stimulation. Left: Each point corresponds to one neuron. Right: Average reward discriminability (AUROC) for control and D2R stimulation (20 neurons, *p* = 0.003). Data are presented as mean values +/- SEM. **G** Reward discriminability (AUROC) in the non-match test period compared between control and D1R stimulation. Left: Each point corresponds to one neuron. Right: Average reward discriminability (AUROC) for control and D1R stimulation (25 neurons, *p* = 0.81). Data are presented as mean values +/- SEM. **H** Reward discriminability (AUROC) in the non-match test period compared between control and D2R stimulation. Left: Each point corresponds to one neuron. Right: Average reward discriminability (AUROC) for control and D2R stimulation (27 neurons, *p* = 0.046). Data are presented as mean values +/- SEM. n.s. not significant, **$p < 0.01$, *$p < 0.05$, signed rank test.

findings, future studies are needed to corroborate D1R modulation of phasic cue responses. In the delay period, however, D1R activation systematically decreased neuronal representations of reward expectations, along with population decoding performance of reward size. This finding was unexpected, given that the same dose of D1R stimulation increased spatial[30] and feature[33] working memory signals, and representations of visual samples and abstract behavioral rules[32]. This result suggests that neuronal networks representing reward signals might be modulated distinctly from networks representing cognitive signals, which are also modulated by D1Rs. A sub-process-specific dopamine modulation could be the result of tailored dopamine input, e.g., independent dopamine signals that carry reward or cognitive information[44,56]. An impaired integration of reward and cognitive signals could underlie previous reports that blocking prefrontal D1Rs impairs association learning and associated neural signals[39], and modulates attentional processing[41]. Mechanistically, D1-mediated inhibition might be realized by increasing inhibitory postsynaptic currents (IPSCs) in prefrontal pyramidal cells[57,58]. However, D1Rs have also been shown to increase NMDA-evoked responses[59,60], possibly contributing to the D1-mediated enhancement of cognitive signals[32,61], suggesting that the present findings are dominated by inhibitory effects.

During later stages of the task, including the sample delay and test periods, we did not observe consistent D1R modulation of reward expectancy signals. However, D1R stimulation increased neuronal responses to reward size during reward delivery, despite the observed overall inhibitory effects. This differential modulation of reward and reward expectation across task epochs indicates that D1R modulation of prefrontal networks is tailored: dlPFC sub-populations with dedicated functional roles might be differentially regulated by prefrontal D1Rs. For instance, signals related to the cognitive load, or cognitive costs, during the delay period might be impaired by D1R stimulation, whereas signals related to reward delivery might be enhanced by D1R stimulation. These results suggest that reward responses and reward expectation signals are realized by dissociable prefrontal circuits with distinct dopaminergic regulation, which could be realized by distinct dopamine neuron populations[49].

D2R stimulation, on the other hand, improved neuronal reward expectancy coding in both cue and delay periods, as well as reward responses, and increased decoding performance of reward size in several task periods. These results suggest that D2Rs could play a prominent role in regulating prefrontal reward and reward expectancy circuits. We have previously found that D2Rs modulate a variety of prefrontal signals, such as feature-based working memory[36] and the representation of abstract behavioral rules[32]. Furthermore, D2Rs modulate cognitive flexibility; blocking prefrontal D2Rs impairs learning of new association rules in primates[62] and impairs rodents in shifting between different response strategies[63]. Thus, D2Rs might contribute to the integration of a variety of prefrontal signals carrying both information about rewards and information relevant for executive control, such as working memory, associations, and rules. In addition, D2Rs are likely involved in the behavioral output, as they modulate saccade signals in dlPFC[35] and influence saccadic target selection in the frontal eye field[41]. In general, D2Rs slightly increased the neurons' spontaneous activity, as reported previously[32,35,64]. Mechanistically, excitatory D2R effects might be mediated by decreasing GABAergic responses in pyramidal cells[57]. At the same time, D2Rs have also been shown to increase interneurons' excitability[65]. D2R stimulation decreased reward size decoding in the cue delay period after excluding reward expectancy neurons, i.e., strongly tuned neurons. This suggests that D2R augment a recurrent winner-take-all neural circuit, in which the tuning of strongly tuned neurons is increased, while the coding of weakly tuned neurons is decreased. Together, these mechanisms might induce an increase in neuronal selectivity during delay periods that relies on recurrent neural circuits,

as suggested by computational modeling[36]. Thus, a general mechanism such as in increase in neuronal gain might explain D2R modulation of working memory, abstract, and reward signals in dlPFC[53].

The phasic discharges of dopaminergic midbrain neurons leading to a sudden and brief release of dopamine are ideally suited to explain the reward prediction error[24,66] and gating signals to dlPFC[26,67,68]. However, during temporal gaps between sensory input and motor output in delayed response or similar tasks, dopamine neurons are typically less active[24,69,70] (see ref. 71 for dopamine responses in delay period in other tasks). The direct activity of dopamine neurons can thus not explain the observed dopaminergic reward expectation effects over the prolonged working memory task. However, three related aspects could play a role. First, dopamine in frontal cortex is not metabolized instantaneously but its levels often last for seconds or minutes[72,73]. Second, release of dopamine is not only caused by the discharges of dopamine neurons; dopamine can also be released at dopamine neurons' terminal endings by local interactions[74]. Third, the differential receptor affinities and distributions across neuron types and cortical layers could enable different time course of dopamine effects in dlPFC[53,75,76]. These mechanisms enable dopamine to differentially act on a long timescale such as the prolonged reward expectation phases used in the current study, even in the absence of transiently activated dopamine neurons. Control and measurement of local dopamine concentrations in dlPFC during cognitive control tasks will be required to differentiate between these possibilities.

We have used dopamine receptor agonists to probe which neural processes are modulated by dopamine receptor activation. While this approach can directly relate dopamine receptor activation with its consequence on neuronal activity and tuning, these experiments do not directly demonstrate to what degree endogenous dopamine release in the dlPFC modulates reward expectancy coding. Further experiments using antagonists or manipulation of dopamine release will be necessary to determine how dopamine signaling affects these processes.

In sum, we found that both D1R and D2R stimulation exerts tailored control of reward expectancy signals in dlPFC. Our results suggest that dopamine informs or modulates dlPFC networks about upcoming rewards and provides the motivational signals that enable successful cognitive control. Vulnerabilities of the dopamine system and cognitive control might therefore partially arise by impairing the integration of the value of future goals with current cognitive demands.

## Methods

### Animals and surgical procedures

Two male rhesus monkeys (*Macaca mulatta*), age 6 and 7 years, were implanted with a titanium head post and one recording chamber centered over the principal sulcus of the dorsolateral prefrontal cortex (dlPFC), anterior to the frontal eye fields (right hemispheres in both monkeys). Surgery was conducted using aseptic techniques under general anesthesia. Structural magnetic resonance imaging was performed before implantation to locate anatomical landmarks. All experimental procedures were in accordance with the guidelines for animal experimentation approved by the national authority, the Regierungspräsidium Tübingen, Germany.

### Task

Monkeys learned to perform a working memory task, in which we manipulated reward expectation by cueing the amount of reward for correct choices in each trial. Monkeys initiated a trial by grasping a lever and maintaining central fixation on a screen. After a pure fixation period (500 ms), a reward cue (300 ms) cued the reward size the monkeys would get for a correct choice at the end of trial (large or small, animal-specific reward amount see below). The reward cue was followed by a delay period (reward cue delay period, 1000 ms) without

visual cues. Then, a visual sample stimulus was presented (600 ms) that monkeys had to memorize during the subsequent delay period (sample delay period, 1000 ms). After the delay period, following a match-to-sample task design, a test stimulus was shown, which was either the same visual item as presented during the sample period (match trial, 50% of trials) or a different visual item (non-match trial, 50% of trials). To make a correct choice, monkeys were required to release a lever during test 1 only if the same matching stimulus appeared and to keep holding the lever for another 1000 ms if a non-matching stimulus appeared, which was followed by the test 2 phase, which always showed a matching stimulus and during which the monkeys had to release the lever (1000 ms). Thus, only test 1 required a decision; test 2 was used so that a behavioral response was required in each trial, ensuring that the monkeys were paying attention during all trials. Monkeys received a water reward for a correct choice with the amount of water determined by the reward cue shown in the beginning of each trial, corresponding to the reward size. We used two reward sizes (small and large) and two reward cue sets (shape and color), i.e., 4 different reward cues in total. In the color cue set, a red square indicated a large reward (monkey 1 0.8 ml, monkey 2 1.0 ml) and a blue square a small reward (monkey 1 0.3 ml, monkey 2 0.2 ml). Reward sizes ware adjusted so that both monkeys showed comparable behavioral performance and effects of cued reward size difference. In the shape set, a gray annulus indicated a large reward, and a gray cross indicated a small reward (same reward amount as in the color conditions). In each session, we used three new different, randomly selected visual items (downloaded from flickr) as sample stimuli. Each sample stimulus served also as non-matching stimulus and vice versa. Trials were pseudo-randomized and balanced across all relevant features (reward size, reward cue set, sample stimulus, match- and non-match-trial). Incorrect trials or trials with broken fixation were repeated at the end of 48-trial blocks to ensure subsequent neural recordings included correct trials balanced across all conditions. Monkeys had to keep their gaze within 1.75° of the fixation point from the fixation interval up to the lever release indicating their choice (monitored with an infrared eye-tracking system; ISCAN, Burlington, MA). If eye fixation was broken during the trial, the trial was aborted followed by a time-out (1000 ms) and counted as a break trial for behavioral analysis.

## Electrophysiology and iontophoresis
Extracellular single-unit recording and iontophoretic drug application was performed as described previously[25,32]. In each recording session, up to three custom-made tungsten-in-glass electrodes flanked by two pipettes each were inserted transdermally using a modified electrical microdrive (NAN Instruments). Single neurons were recorded at random; no attempt was made to preselect the neurons to any task-related activity or based on drug effects. Signal acquisition, amplification, filtering, and digitalization were accomplished with the MAP system (Plexon). Waveform separation was performed offline (Offline Sorter; Plexon).

Drugs were applied iontophoretically (MVCS iontophoresis system; npi electronic) using custom-made tungsten-in-glass electrodes flanked by two pipettes each[25,32,77]. Electrode impedance and pipette resistance were measured after each recording session. Electrode impedances were 0.8–3 MΩ (measured at 500 Hz; Omega Tip Z; World Precision Instruments). Pipette resistances depended on the pipette opening diameter, drug, and solvent used. Typical resistances were 15–50 MΩ (full range, 12–160 MΩ). As in previous experiments[25,32], we used retention currents of −7 nA to hold the drugs in the pipette during control conditions. The ejection current for SKF81297 (10 mM in double-distilled water, pH 4.0 with HCl; Sigma-Aldrich) was +15 nA, and the ejection current for quinpirole (10 mM in double-distilled water, pH 4.0 with HCl; Sigma-Aldrich) was +40 nA. We did not investigate dosage effects and chose ejection currents to match the values reported to be maximally effective, i.e., in the peak range of the

'inverted-U function'[30,32,35]. One pipette per electrode was filled with drug solution (either SKF81297 or quinpirole), and the other always contained 0.9% NaCl. In each recording session, control conditions using the retention current alternated with drug conditions using the ejection current. Drugs were applied continuously for about 12 min (drug conditions), depending on the number of trials completed correctly by the animal. Each control or drug application block consisted of 72 correct trials to yield sufficient trials for analysis. The first block (12 min) was always the control condition.

A recording session typically comprised 2 control and 2 drug sessions. Most neurons (75%) were recorded including the first control session. Given that iontophoretic drug application is fast and can quickly modulate neuronal firing properties[25], we did not exclude data at the current switching points. We repeated the main analyses from Figs. 3–6 excluding trials within a 120 s wash out period after switching from a drug to a control block, which showed the same general effects (Reward cue, D1: $\Delta$ AUROC = +0.0051 ± 0.0098, $n$ = 24, $p$ = 0.6; D2: $\Delta$ AUROC = +0.25 ± 0.012, $n$ = 27, $p$ = 0.075. Cue Delay, D1: $\Delta$ AUROC = −0.032 ± 0.014, $n$ = 29, $p$ = 0.013; D2: $\Delta$ AUROC = +0.021 ± 0.0091, $n$ = 41, $p$ = 0.007. Sample delay, D1: $\Delta$ AUROC = −0.0061 ± 0.013, $n$ = 31, $p$ = 0.9; D2: $\Delta$ AUROC = +0.023 ± 0.0088, $n$ = 41, $p$ = 0.008. Reward response, D1: $\Delta$ AUROC = +0.034 ± 0.016, $n$ = 16, $p$ = 0.068; D2: $\Delta$ AUROC = +0.051 ± 0.017, $n$ = 18, $p$ = 0.010).

## Data analyses
**Reward expectancy neurons.** All well-isolated recorded single units with a baseline spike rate above 0.5 spikes/s (determined in the 500 ms fixation period preceding sample presentation) and at least 12 trials in each reward size, reward cue set, and iontophoretic drug application condition (i.e., at least 96 trials in total) entered the analyses ($n$ = 256 neurons). Each neuron was recorded for a median of 124 correct trials (median 62 trials for the small reward and 62 trials for the large reward). Neurons were not included based on drug effects. We used analysis of variance (ANOVA) for each neuron to determine if a neuron's response was correlated with reward expectancy using spike rates in four different, non-overlapping task epochs. The first analysis period, the *cue period*, was defined using a 400 ms window beginning 100 ms after reward cue onset. The second analysis period, the *cue delay period*, was defined using a 900 ms window beginning 200 ms after reward cue offset. The third analysis period, the *sample delay period*, was defined using a 900 ms window beginning 200 ms after memory sample offset. The fourth analysis period, the *non-match test period*, was defined using a 900 ms window beginning 200 ms after presenting a non-matching test stimulus. In a fifth analysis period, the *reward period*, we determined whether a neuron's response correlated with reward size after reward delivery, using a 500 ms window starting 100 ms after reward delivery in correct match trials. Main factors for each ANOVA were reward size (large/small), reward cue set (color set/ shape set) and iontophoretic drug condition (control/drug application), including interaction terms. For the sample delay period, we added a main factors and interaction terms for the memory sample (pair-wise contrast coding). We labelled a neuron as selective for reward expectancy (or selective for reward size in the reward period), if it showed a significant main effect of reward size ($p$ < 0.05) and no significant main effect of reward cue set ($p$ > 0.05) and no interaction between reward size and reward cue set ($p$ > 0.05). Thus, we isolated neurons encoding reward expectancy signals (cue, cue delay, sample delay periods) or responding to reward size (reward period). Only few reward expectancy neurons also encoded the sample stimulus in the same delay phase (10/79 in the cue delay period; 11/74 in the sample delay period).

**Single-cell and population responses.** For plotting single-cell spike density histograms, the average firing rates in trials with one of the four different reward size cues (correct trials only) were smoothed with a

Gaussian kernel (bin width of 200 ms, steps of 1 ms) for visual presentation only. For the population responses, trials with reward size cues signifying the same reward size were pooled. A neuron's preferred reward size was defined as the reward size (large or small) yielding the higher average spike rate in the analysis windows used for the ANOVAs. The nonpreferred reward size was defined as the reward size resulting in lower average spike rate. Neuronal activity was normalized by subtracting the mean baseline firing rate (500 ms fixation period preceding reward cue presentation) in the control condition and dividing by the standard deviation of the baseline firing rates in the control condition. For population histograms, normalized activity was averaged and smoothed with a Gaussian kernel (width of 200 ms, step of 1 ms) for visual presentation only.

**Reward discriminability encoding.** We quantified each neuron's discriminability for reward size (small vs. large) using receiver operating characteristic (ROC) analysis derived from signal detection theory[78]. The area under the ROC curve (AUROC) is a nonparametric measure of the discriminability of two distributions. It denotes the probability with which an ideal observer can tell apart a meaningful signal from a noisy background. Values of 0.5 indicate no separation, and values of 1 signal perfect discriminability. We chose the firing rates for the large reward size as signal distribution, and the firing rates for the small reward size as noise distribution; by this convention, AUROC-values larger than 0.5 indicated discriminability for large-reward expectancy neurons, and values smaller than 0.5 signal discriminability for small-reward preferring expectancy neurons. The AUROC takes into account both the difference between distribution means as well as their widths and is therefore a suitable indicator of signal quality[79]. We calculated the AUROC for each neuron using the spike rate distributions of the preferred and the nonpreferred reward size in the same analysis windows used for the ANOVAs. Pair-wise non-parametric statistical tests were used as indicated to compare selectivity between control and drug phases.

To explore temporal reward expectancy throughout the task, we calculated AUROCs between spike rate distributions in all large-reward and small-reward trials using a sliding 150 ms window (10 ms steps, control condition only) separately for all reward-expectancy-selective neurons (neurons selective in either task period, see above). Neurons were grouped into overall large- or small-reward-preferring based on their firing rate average across the entire trials (fixation period to match or nonmatch onset). For visualization purposes, we estimated each neuron's latency of reward response as the first time point when the AUROC was higher than three standard deviations of baseline AUROCs (fixation period before reward cue stimulus). If no latency could be determined, neurons were grouped together with the largest latency.

To explore typical temporal selectivity profiles for reward expectancy, we used principal component analyses (PCA) of the z-scored selectivity matrix including all neurons (AUROC entries for each neuron and timepoint, $n$Neurons x $n$Timepoints with $n$Neurons = 245 and $n$Timepoints = 340, control condition only). Neuron loadings were typically sparse for the first 3 temporal components (sorted by their eigenvalues, on average 60% of a neuron's loading between the first 3 components fell onto a single component), indicating that reward-selective neurons followed one of the temporal time courses visualized by the first 3 temporal components. These time courses supported the inclusion of 3 main analysis windows for reward expectancy encoding (reward cue, cue delay, sample delay).

**Decoding analysis.** We used decoding analyses to quantify the degree to which population activity predicted reward size in each analysis window (cue, cue delay, sample delay periods). We constructed a set of pseudo-simultaneous recorded trials (35 trials per reward size and iontophoresis condition, corresponding to the minimum number of

trials recorded in 90% of all neurons) by randomly sub-sampling large-reward and small-reward trials (separately for control and drug conditions) and calculating the average spike rate in the respective analysis window, yielding a population activity vector for each trial. We then trained a linear decoder (using a support vector machine classifier) to predict the binary class "reward size" (small or large) of the test trial using leave-one-out cross-validation, iterating over trials such that each trial served as the test set once. The decoder's performance was then given by the percentage of correctly classified trials. Standard errors of the decoder's performance across classes (confusion matrices), and statistically significant performance differences between control and drug conditions were estimated using $n = 1000$ bootstrap samples ($p = 0.05$, two-sided). To estimate the number of neurons required for decoder performance, we repeated the decoder analysis by randomly subsampling neurons.

### Reporting summary
Further information on research design is available in the Nature Portfolio Reporting Summary linked to this article.

## Data availability
The data that support the findings of this study are available from the corresponding author upon request. The data can only be made available from the authors on request because the data await further analysis. Source data are provided with this paper.

## Code availability
The code that support the findings of this study is available from the corresponding author upon request. Only customarily available code was used.

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

## Acknowledgements

This work was supported by grants from the German Research Foundation (DFG) to A.N. (NI 618/5-2 and NI 618/13-1) and T.O. (OT 562-2/1).

## Author contributions

T.O. designed and performed experiments, analyzed data, and wrote the paper. A.M.S. performed experiments. A.N. designed experiments, evaluated data, wrote the paper, and supervised the study.

## Funding

## Competing interests

The authors declare no competing interests.
