## [Peer Review File · Nature Communications]

Dopamine receptor activation regulates reward expectancy signals during cognitive control in primate prefrontal neuronsREVIEWER COMMENTS

Reviewer #1 (Remarks to the Author):

Review Nature Communications

Ott et al. examine how stimulation of dopamine receptor subtypes affects reward representation in the activity of neurons in the primate dorsolateral prefrontal cortex. Several studies have examined how reward is represented in different prefrontal areas. Previous studies in the DLPFC have delineated the modulation of neuronal activity with reward value and additionally, how reward value modulation affects prefrontal representation of stimuli in working memory.

Dopamine neuromodulation is intimately involved in reward processing and there is extensive literature documenting dopamine modulation of prefrontal cortical neurons during working memory and other tasks involving attention and learning of associations. Dopamine depletion in DLPFC has profound effects on the performance of these tasks. Dopamine neurons fire during the encoding of stimuli in to working memory and phasic and tonic dopamine release modulated PFC activity through volume transmission and signal transduction at D1 and D2 receptors.

Previous studies have examined the contribution of these receptor subtypes in PFC to stimulus encoding, representation of stimuli in working memory, and motor activity in PFC. However, the influence of these receptors on the encoding of reward value and motivational context in PFC is unknown. This study makes an important contribution to our understanding of this less understood, yet crucial, aspect of dopamine modulation of PFC.

The authors demonstrate that dopamine D1 and D2 receptors differentially modulate the encoding of reward expectancy at various points in the task trials. They find that D1 receptors degrade the representation of stimuli that represent the expected reward in working memory, while enhancing the representation of reward after the delivery of the same. In contrast, D2 receptors uniformly augment reward expectancy through the course of the trial.

Review comments:

1. Differential activity in the high and low rewarded trials in this study could encode a computation of relative value. Alternatively, such activity could be a consequence of differential motivation due to the difference in expected reward. Understanding dopamine modulation of PFC in the context of distinguishing reward value-related variables versus motivational context is a potentially interesting subject. Roesch and Olson (Science, 2004) distinguished between these possibilities with a task involving cues associated with different positive (fluid reward) and negative reinforcement (differential time-outs) to compare activity in premotor cortex and OFC. They found that premotor neurons increased their firing when reward was high and when the penalty timeout was long. In contrast, OFC neurons only modulated with the reward value. They interpreted this to imply that premotor neurons encoded the motivational context, since both high reward vs low reward and high penalty vs low penalty would presumably provide high motivation to select a particular action (getting higher reward or avoiding higher penalty). Other studies (e.g., Leon and Shadlen, Neuron, 1999, Matsumoto and Takada, Neuron, 2013, Kobayashi et al., Neuron, 2006) have described reward value modulation in DLPFC to be stronger during working memory performance rather than in simpler task contexts.

While the task employed here is different, it is worth noting that the structure of the task in the non-match trials can be interesting in this context. In the nonmatch trial, the monkey must forego releasing the lever and wait for subsequent presentation of the match stimulus to release the lever to get reward. In this epoch, between the presentation of the nonmatch and the subsequent match presentation and lever release, the subject no longer has working memory load and only has to release the lever after the invariable subsequent presentation of the match stimulus to get the reward. A question one could ask is whether reward modulation in this epoch of the non match trial was similar to the modulation seen in the earlier epochs of the same trial. If reward anticipation signals observed here were encoding merely the value of the reward, one would expect that the modulation in the last epoch of then nonmatch task would track modulation in the previous epochs of the same trial and be similar to the delayed match trials as well. In contrast, if the reward modulation of the activity in this epoch is attenuated, this might indicate that the reward effect is due to something else, perhaps motivational context or cognitive load at that point in the trial. The

authors may wish to examine this.

2. Related to 1), the authors find intriguing effects of D1R on reward-related activity in the post-reward epoch that contrast with its effects in other epochs. D1R stimulation enhanced reward-related activity differences in the post-reward epoch but was detrimental to the same activity in the reward-cue delay epoch.

Firstly, did this analysis also include the (later) reward in the non-match trials? Secondly, it is conceivable, within the framework of mixed selectivity of PFC neurons, that the reward-differential activity in the post-reward epoch is functionally and mechanistically different from earlier in the trial: differential activity during the trial might be a consequence of motivational or task/effort context, while activity after the reward encodes the value of the reward just received? If this is the case, opponent D1R effects on the earlier reward cue-delay epoch versus post-reward could potentially indicate that D1 receptors differentially modulate reward value and motivation/cognitive load related variables. This possibility may merit brief discussion. Consistent with this, Matsumoto and Takada (Neuron, 2013) found dopamine neurons that encode task/cognitive context and reward prediction signals in different areas in the midbrain dopaminergic system.

3. The authors find that neurons with higher activity for small reward during the working memory trial are more numerous than those with higher activity for higher reward. Was this also true of the population with differential reward activity in the post-reward period? Cursorily, the AUROC plots in Figure 7 seem to indicate that this is not the case. However, in the interest of consistency, it would be expeditious to include the numbers in Table 1, where counts for the other epochs are indicated.

4. The decoding analysis found an interesting apparent discrepancy between the effects of D1R in the reward-cue period on reward discriminability with the ROC analysis and with decoding analysis. The authors found that the increase in reward size decoding was driven by neurons that were not included in the reward discriminability analysis (presumably no effect of reward size in the ANOVA). This implies that D1R augmented the reward tuning of weakly-tuned reward activity of PFC neurons. The authors show that this trend exists in their population excluding reward-tuned neurons curated from the ANOVA. Quinpirole (D2)

results did not vary between the AUROC analysis and the decoding analysis. Did quinpirole also consistently affect weakly-tuned neurons that did not reach significance in the ANOVA-based classification in any of the epochs?

5. The authors report an intriguing finding that more of the reward-tuned neurons preferred the small reward than the big reward. In the discussion they mentioned that this is consistent with the findings of Kennerley and Wallis (J. Neurosci, 2009). However, that study indicates that spatial selectivity was enhanced in the small reward population in DLPFC in one configuration of the task (Fig 6 and 7 of that study). It is not apparent from that study that the number of small reward preferring neurons is larger in PFC.

However, Kobayashi et al. (J. Neurophysiol, 87: 1488 –1498, 2002) recorded DLPFC neurons while monkeys performed a delayed saccade task with an asymmetric reward schedule (one stimulus was rewarded, the other was not) with blockwise reversal of stimulus contingencies. They found neurons that encoded the stimulus location, reward size (R) and stimulus-reward neurons encoding both stimulus location and reward (SR). In the R and SR populations, they distinguished between R+ and R- (preferring reward and no reward, respectively) and SR+ and SR- (again preferring reward and no reward, respectively).

Interestingly, they found that R- type neurons were more numerous than R+ type neurons in DLPFC, but this was not true for SR+ and SR- neurons (Table 2 in that study). They also argue that the previous study of Leon and Shadlen (Neuron, 1999) did not find such low reward neurons because, the task in that study distinguished low and high reward versus reward and no reward. This study uses a similar task paradigm, like Leon and Shadlen's study, with low and high reward. However, this study finds more small reward preferring neurons, in contrast to Leon and Shadlen. The authors may wish to highlight this.

Further, if the same is true in the dataset in this manuscript as was the case for Kobayashi et al., it would imply a plurality of the neurons that showed reward selectivity should not possess match to sample stimulus-related selectivity. The authors may wish to examine how many neurons in their dataset showed both sample stimulus selectivity in addition to reward selectivity, data which may already be present in the ANOVA analysis for the DMS delay epoch. In any case, the authors may wish to discuss their finding of greater numbers of "low reward" neurons in this context.

6. The authors have previously reported the effects of dopamine receptors on sample

selectivity in a numerosity delayed match to sample task with similar task structure (Ott et al, Neuron, 2014), wherein D1 receptors enhanced numerosity encoding in the sample epoch, while D2R had no effect. One question that arises is whether dopamine receptor modulation of reward anticipation activity reported here in turn had effects on sample encoding activity. Since a stimulus in this task can be the match and the non-match, it would be interesting to see if the representation of the stimulus in these contexts is differentially affected in large and small-reward preferring neurons respectively (as observed by Kobayashi et al). Alternatively, reward modulation may only affect the working memory representation of the sample or have no influence at all on sample representation all. However, I completely understand that this may be outside the scope of this manuscript or the authors may be intending a follow-up publication to examine this aspect of their dataset.

Reviewer #2 (Remarks to the Author):

This manuscript by Ott, Stein, and Nieder reports the results of a match-to-sample working memory task in nonhuman primates. Although the backbone of the study is a standard working-memory task, the authors added a new variable, which makes it more innovative. The authors manipulated the animals' reward expectancy, as evidenced in the spiking activity of prefrontal neurons, and assessed the contribution of distinct types of dopamine neurons in driving this reward expectancy. They found that pharmacological activation of D2-type dopamine receptors in PFC enhanced the reward expectancy signals at several trial epochs, while activation of D1-type receptors had mixed effects.

My comments:

- The main drawback of the study is that the authors show what effects dopamine could, in principle, have in these neural signals. They do not show data to support that, indeed, dopaminergic signaling in PFC is responsible for the reward expectancy in PFC neurons. The authors should (1) acknowledge this drawback in the Discussion and (2) edit the text accordingly (e.g., line 356 should change to "These results suggest that D2Rs may play prominent role..."). Similarly, the title of the manuscript should change to something along

of the lines of “Dopamine receptors can potentially regulate...”

- The differential effect of high- vs. low-reward expectancy on fixation breaks indicates that the two conditions included different number of trials. The authors should mention in the Results or Methods sections how different was the average number of trials (or trial blocks) successfully completed in each reward condition and, if warranted, what measures they took to ensure the reported effects had not arisen from these differences between number of trials (e.g., did they use random resampling methods to match the trial numbers?)

- It is not clear how many blocks of drug and control conditions the authors repeated in each animal. Did they collect any data in the Control condition following the Drug condition? If so, how did they ensure there was no residual effect of the drug they had previously applied? Did they examine the time course of the drug washout in the neural activity?

- This is not necessary (and may be beyond the scope of this study), but I would be interested to see how the neural signals (and the receptor-type effects) change when the animal transitions from a low-reward condition (possibly after a few consecutive trials) to a high-reward condition and vice versa. Do the authors find PFC neurons that signal the change independently of the direction of change, and which type of dopamine receptor can modulate this signal?

Evan Antzoulatos

Revision of manuscript NCOMMS-23-17343

Reply to reviewers

Reviewer #1:

Ott et al. examine how stimulation of dopamine receptor subtypes affects reward representation in the activity of neurons in the primate dorsolateral prefrontal cortex. Several studies have examined how reward is represented in different prefrontal areas. Previous studies in the DLPFC have delineated the modulation of neuronal activity with reward value and additionally, how reward value modulation affects prefrontal representation of stimuli in working memory.

Dopamine neuromodulation is intimately involved in reward processing and there is extensive literature documenting dopamine modulation of prefrontal cortical neurons during working memory and other tasks involving attention and learning of associations. Dopamine depletion in DLPFC has profound effects on the performance of these tasks. Dopamine neurons fire during the encoding of stimuli in to working memory and phasic and tonic dopamine release modulated PFC activity through volume transmission and signal transduction at D1 and D2 receptors.

Previous studies have examined the contribution of these receptor subtypes in PFC to stimulus encoding, representation of stimuli in working memory, and motor activity in PFC. However, the influence of these receptors on the encoding of reward value and motivational context in PFC is unknown. This study makes an important contribution to our understanding of this less understood, yet crucial, aspect of dopamine modulation of PFC.

The authors demonstrate that dopamine D1 and D2 receptors differentially modulate the encoding of reward expectancy at various points in the task trials. They find that D1 receptors degrade the representation of stimuli that represent the expected reward in working memory, while enhancing the representation of reward after the delivery of the same. In contrast, D2 receptors uniformly augment reward expectancy through the course of the trial.

Review comments:

1. Differential activity in the high and low rewarded trials in this study could encode a computation of relative value. Alternatively, such activity could be a consequence of differential motivation due to the difference in expected reward. Understanding dopamine modulation of PFC in the context of distinguishing reward value-related variables versus motivational context is a potentially interesting subject. Roesch and Olson (Science, 2004) distinguished between these possibilities with a task involving cues associated with different positive (fluid reward) and negative reinforcement (differential time-outs) to compare activity in premotor cortex and OFC. They found that premotor neurons increased their firing when reward was high and when the penalty timeout was long. In contrast, OFC neurons only modulated with the reward value. They interpreted this to imply that premotor neurons encoded the motivational context, since both high reward vs low reward and high penalty vs low penalty would presumably provide high motivation to select a particular action (getting higher reward or avoiding higher penalty). Other studies (e.g., Leon and Shadlen, Neuron, 1999, Matsumoto and Takada, Neuron, 2013, Kobayashi et al., Neuron, 2006) have described

reward value modulation in DLPFC to be stronger during working memory performance rather than in simpler task contexts.

While the task employed here is different, it is worth noting that the structure of the task in the non-match trials can be interesting in this context. In the nonmatch trial, the monkey must forego releasing the lever and wait for subsequent presentation of the match stimulus to release the lever to get reward. In this epoch, between the presentation of the nonmatch and the subsequent match presentation and lever release, the subject no longer has working memory load and only has to release the lever after the invariable subsequent presentation of the match stimulus to get the reward. A question one could ask is whether reward modulation in this epoch of the non match trial was similar to the modulation seen in the earlier epochs of the same trial. If reward anticipation signals observed here were encoding merely the value of the reward, one would expect that the modulation in the last epoch of then nonmatch task would track modulation in the previous epochs of the same trial and be similar to the delayed match trials as well. In contrast, if the reward modulation of the activity in this epoch is attenuated, this might indicate that the reward effect is due to something else, perhaps motivational context or cognitive load at that point in the trial. The authors may wish to examine this.

We thank the reviewer for these excellent suggestions, which served as the basis for a novel set of analyses (novel panels **Figure 7G,H**) further strengthening the conclusions of our study. As the reviewer notes, our task design does not allow for a strict separation between reward anticipation dissociated from motivation as in the study by Roesch and Olson (2004). We therefore do not make strong conclusions about interpreting neural signals related to only one of these states and use the more general term 'reward expectancy', which might contribute to differential degrees of motivation between low and large reward conditions. Nevertheless, following the reviewer's suggestion, we have added an analysis of dopamine receptor modulation of reward expectancy signals in the non-match period, a 1-second period after presentation of the non-match stimulus during which the monkeys were required to keep holding the lever until match presentation. As the reviewer points out, during this task epoch, the cognitive load is reduced, since the monkeys do not need to remember the stimulus features. While we cannot fully rule out motivational factors, this task epoch relies less on motivation for task performance, given that the only remaining task is holding the lever for 1s and no working memory is required. We observed a similar modulation of reward expectancy signals by D1Rs and D2Rs as in the cue and delay periods: While D1Rs lacked a consistent modulation, D2Rs significantly increased reward discriminability between small and large reward (**Figure 7G,H**). We further observed similar numbers of reward-size-selective neurons, again, with a larger number of small-reward-preferring neurons (**extended Table 1**). Together these results are consistent with a continuous modulation of reward expectancy signals throughout the delay periods, in particular for D2Rs, and therefore suggest that dopamine receptors modulate signals related to the relative value between small and large reward rather than signals related to cognitive load or motivation. We report the new findings in the Method (**line 542-544**), the Results (**lines 295-308**), and the Discussion (**lines 334-343**) and added novel subpanels **Figure 7G,H**

2. Related to 1), the authors find intriguing effects of D1R on reward-related activity in the post-reward epoch that contrast with its effects in other epochs. D1R stimulation enhanced reward-related activity differences in the post-reward epoch but was detrimental to the same activity in the reward-cue delay epoch. Firstly, did this analysis also include the (later) reward in the non-match trials? Secondly, it is conceivable, within the framework of mixed selectivity

of PFC neurons, that the reward-differential activity in the post-reward epoch is functionally and mechanistically different from earlier in the trial: differential activity during the trial might be a consequence of motivational or task/effort context, while activity after the reward encodes the value of the reward just received? If this is the case, opponent D1R effects on the earlier reward cue-delay epoch versus post-reward could potentially indicate that D1 receptors differentially modulate reward value and motivation/cognitive load related variables. This possibility may merit brief discussion. Consistent with this, Matsumoto and Takada (Neuron,2013) found dopamine neurons that encode task/cognitive context and reward prediction signals in different areas in the midbrain dopaminergic system.

We are grateful to the reviewer for these insightful comments. In agreement with the reviewer, we would argue that pre- and post-reward responses are functionally different. These functions might therefore be realized by dedicated PFC neural populations susceptible to tailed D1R modulation. While at this stage this remains speculative, our D1R data points towards this direction. We've added a discussion segment to highlight these ideas (**lines 373-383**). The reward analysis only included match-trial reward responses (clarified in methods).

3. The authors find that neurons with higher activity for small reward during the working memory trial are more numerous than those with higher activity for higher reward. Was this also true of the population with differential reward activity in the post-reward period? cursorily, the AUROC plots in Figure 7 seem to indicate that this is not the case. However, in the interest of consistency, it would be expeditious to include the numbers in Table 1, where counts for the other epochs are indicated.

In the post-reward period, a higher absolute number of neurons increased their activity after the monkeys received a large reward vs. a small reward, although the proportions of these neurons were not significantly different (Chi-square-test). In the newly added non-match test period (cf. comment #1), we observed more neurons that preferred the upcoming small reward. We have added these numbers to **Table 1** and report the statistics in the respective results sections (**lines 279-281 and 301-302**).

4. The decoding analysis found an interesting apparent discrepancy between the effects of D1R in the reward-cue period on reward discriminability with the ROC analysis and with decoding analysis. The authors found that the increase in reward size decoding was driven by neurons that were not included in the reward discriminability analysis (presumably no effect of reward size in the ANOVA). This implies that D1R augmented the reward tuning of weakly-tuned reward activity of PFC neurons. The authors show that this trend exists in their population excluding reward-tuned neurons curated from the ANOVA. Quinpirole (D2) results did not vary between the AUROC analysis and the decoding analysis. Did quinpirole also consistently affect weakly-tuned neurons that did not reach significance in the ANOVA-based classification in any of the epochs?

Even though D2R stimulation increased reward discriminability as measured by the AUROC in all task periods, and decoding performance increased in a subset of these periods, decoding performance after excluding significant reward expectancy neurons did not significantly increase in any of the task epochs. Only in the reward cue delay period did D2R stimulation decreased the decoder's performance after excluding selective neurons (Δ performance = -0.16 + 0.04, $p = 0.042$), i.e., surprisingly, the opposite effect as for the overall population and for the reward discriminability analysis. This suggests that the D2R-mediated coding increase were primarily driven by the population of strongly selective neurons, while weakly tuned

neurons were either not modulated or their selectivity was decreased. We added these findings in the Results section (**lines 258-267**) and added a discussion of this finding (**lines 401-406**).

5. The authors report an intriguing finding that more of the reward-tuned neurons preferred the small reward than the big reward. In the discussion they mentioned that this is consistent with the findings of Kennerley and Wallis (J. Neurosci, 2009). However, that study indicates that spatial selectivity was enhanced in the small reward population in DLPFC in one configuration of the task (Fig 6 and 7 of that study). It is not apparent from that study that the number of small reward preferring neurons is larger in PFC. However, Kobayashi et al. (J. Neurophysiol, 87: 1488-1498, 2002) recorded DLPFC neurons while monkeys performed a delayed saccade task with an asymmetric reward schedule (one stimulus was rewarded, the other was not) with blockwise reversal of stimulus contingencies. They found neurons that encoded the stimulus location, reward size (R) and stimulus-reward neurons encoding both stimulus location and reward (SR). In the R and SR populations, they distinguished between R+ and R- (preferring reward and no reward, respectively) and SR+ and SR- (again preferring reward and no reward, respectively). Interestingly, they found that R- type neurons were more numerous than R+ type neurons in DLPFC, but this was not true for SR+ and SR- neurons (Table 2 in that study). They also argue that the previous study of Leon and Shadlen (Neuron, 1999) did not find such low reward neurons because, the task in that study distinguished low and high reward versus reward and no reward. This study uses a similar task paradigm, like Leon and Shadlen's study, with low and high reward. However, this study finds more small reward preferring neurons, in contrast to Leon and Shadlen. The authors may wish to highlight this. Further, if the same is true in the dataset in this manuscript as was the case for Kobayashi et al., it would imply a plurality of the neurons that showed reward selectivity should not possess match to sample stimulus-related selectivity. The authors may wish to examine how many neurons in their dataset showed both sample stimulus selectivity in addition to reward selectivity, data which may already be present in the ANOVA analysis for the DMS delay epoch. In any case, the authors may wish to discuss their finding of greater numbers of "low reward" neurons in this context.

We thank the reviewer for their expertise and insight into the relation of our results with previous studies in primate prefrontal cortex investigating the relation between reward size and working memory. We have reworked the Discussion (**lines 322-324**) to more precisely relate to previous studies that found a larger number of no-reward or low-value preferring neurons (such as Kobayashi et al., 2002) and the differences to Leon and Shadlen's (1999) study.

We further agree with the reviewer that the relationship with sample/working memory coding is important, and that neurons may fall into different groups that exhibit differential dopaminergic modulation – a hypothesis that motivated the present study. We have extended our analyses to distinguish between reward expectancy neurons that additionally encode the sample stimulus in the sample delay period. In brief, only a few of our reward expectancy neurons additionally encoded the sample stimulus (10/79 in the cue delay period; 11/74 in the sample delay period). We note that this contrasts with some previous findings, in which SR-type neurons were quite prevalent (as in Kobayashi et al). Thus, our findings primarily pertain to R-type neurons. We repeated the main analyses after excluding neurons that encode the sample after sample presentation and find the same main effects by D1R and D2R stimulation (cue period, D1: $\delta = +0.007 \pm 0.01$, $p = 0.39$; cue period, D2: $\delta = +0.03 \pm 0.01$, $p = 0.043$; cue delay period, D1: $\delta = -0.033 \pm 0.014$, $p = 0.029$; cue delay period, D2:

delta = +0.031 ± 0.009, p = 0.0008). We have added these considerations to the Methods (lines 554-556) and the Discussion (lines 339-343) section. We prefer not to include this control analysis to the final manuscript given that the main results are unchanged and considering the small number of SR-type neurons in our dataset.

6. The authors have previously reported the effects of dopamine receptors on sample selectivity in a numerosity delayed match to sample task with similar task structure (Ott et al, Neuron, 2014), wherein D1 receptors enhanced numerosity encoding in the sample epoch, while D2R had no effect. One question that arises is whether dopamine receptor modulation of reward anticipation activity reported here in turn had effects on sample encoding activity. Since a stimulus in this task can be the match and the non-match, it would be interesting to see if the representation of the stimulus in these contexts is differentially affected in large and small-reward preferring neurons respectively (as observed by Kobayashi et al). Alternatively, reward modulation may only affect the working memory representation of the sample or have no influence at all on sample representation all. However, I completely understand that this may be outside the scope of this manuscript or the authors may be intending a follow-up publication to examine this aspect of their dataset.

We fully agree with the reviewer that these are interesting questions. However, in this study we find relatively few neurons that display an interaction between reward size and sample coding (see previous comment), limiting possible conclusion about their modulation by receptors. We therefore feel these questions are outside the scope of this manuscript and warrant additional follow-up studies.

Reviewer #2:

This manuscript by Ott, Stein, and Nieder reports the results of a match-to-sample working memory task in nonhuman primates. Although the backbone of the study is a standard working-memory task, the authors added a new variable, which makes it more innovative. The authors manipulated the animals' reward expectancy, as evidenced in the spiking activity of prefrontal neurons, and assessed the contribution of distinct types of dopamine neurons in driving this reward expectancy. They found that pharmacological activation of D2-type dopamine receptors in PFC enhanced the reward expectancy signals at several trial epochs, while activation of D1-type receptors had mixed effects.

My comments:

- The main drawback of the study is that the authors show what effects dopamine could, in principle, have in these neural signals. They do not show data to support that, indeed, dopaminergic signaling in PFC is responsible for the reward expectancy in PFC neurons. The authors should (1) acknowledge this drawback in the Discussion and (2) edit the text accordingly (e.g., line 356 should change to "These results suggest that D2Rs may play prominent role"). Similarly, the title of the manuscript should change to something along the lines of "Dopamine receptors can potentially regulate"

We thank the reviewer for pointing out this limitation. We agree that more precise language is helpful and have edited the manuscript accordingly. Specifically, we have added a "Limitations" paragraph at the end of the Discussion highlighting these and additional considerations (lines 430-436). We have also edited several sentences throughout the manuscript, including the sentence mentioned by the reviewer (now line 388), and modified the title.

- The differential effect of high- vs. low-reward expectancy on fixation breaks indicates that the two conditions included different number of trials. The authors should mention in the Results or Methods sections how different was the average number of trials (or trial blocks) successfully completed in each reward condition and, if warranted, what measures they took to ensure the reported effects had not arisen from these differences between number of trials (e.g., did they use random resampling methods to match the trial numbers?)

This is an important consideration. Each neuron was recorded for a median number of 124 correct trials that entered the analyses, with a median of 62 large and 62 small reward trials. This number is balanced because incorrect trials or trials with broken fixation trials were repeated after 1-48 trials to ensure we recorded from all conditions with balanced numbers of correct trials. Thus, the trial numbers and statistical power to detect large- and small-preferring neurons was the same. We have added these numbers and considerations to the Methods section (lines 483-485 and 534-535).

- It is not clear how many blocks of drug and control conditions the authors repeated in each animal. Did they collect any data in the Control condition following the Drug condition? If so, how did they ensure there was no residual effect of the drug they had previously applied? Did they examine the time course of the drug washout in the neural activity?

We alternated between control and drug-application blocks that comprised 72 trials each, and a neuron was typically recorded for 124 correct trials (see previous comment). Each recording session typically consisted of 2 control and 2 drug blocks. Given recording quality was typically best in the beginning of a session, most neurons were recorded during the first control and

drug blocks (75 % of recorded neurons), while occasionally neurons were recorded in multiple control and/or drug application blocks. We have added these details to the Methods section (lines 517-518 and 534-535).

In previous studies using the same methods and drugs, we have extensively characterized the wash-in and wash-out time courses. We found that drugs washed out within a few minutes and reduced substantially within a minute (Ott et al., Neuron, 2014; Jacob et al., J Neuro, 2013). It is further difficult to assess wash out times precisely, since we cannot measure pharmacological concentrations and rely on subtle changes in neural activity. Importantly, an incomplete wash out would lead to missing an effect and not produce a false positive. At the same time, reducing the number of trials when excluding trials during putative wash-out will reduce statistical power and can therefore also contribute to missing effects (depending on the unknown magnitude of residual drug effects), especially since analyses rely on including trials from multiple task conditions. Given these complexities, we chose, as in previous studies, not to exclude trials in putative wash in our wash out periods. Nevertheless, we repeated the main analyses from Figures 3-6 excluding trials within a 120 second wash out period after switching from a drug to a control block. While the number of reward expectancy neurons was decreased due to neurons being excluded from analyses due to insufficient trials in at least one task condition, the main results stayed largely the same.

Reward cue: D1: Δ AUROC = $+0.0051 \pm 0.0098$, n = 24, p = 0.6. D2: Δ AUROC = $+0.25 \pm 0.012$, n = 27, p = 0.075.

Cue Delay: D1: Δ AUROC = -0.032 ± 0.014 , n = 29, p = 0.013. D2: Δ AUROC = $+0.021 \pm 0.0091$, n = 41, p = 0.007.

Sample delay: D1: Δ AUROC = -0.0061 ± 0.013 , n = 31, p = 0.9. D2: Δ AUROC = $+0.023 \pm 0.0088$, n = 41, p = 0.008.

Reward response: D1: Δ AUROC = $+0.034 \pm 0.016$, n = 16, p = 0.068. D2: Δ AUROC = $+0.051 \pm 0.017$, n = 18, p = 0.010.

We included a summary of these results in the Methods section as a control analysis (lines 517-527).

- This is not necessary (and may be beyond the scope of this study), but I would be interested to see how the neural signals (and the receptor-type effects) change when the animal transitions from a low-reward condition (possibly after a few consecutive trials) to a high-reward condition and vice versa. Do the authors find PFC neurons that signal the change independently of the direction of change, and which type of dopamine receptor can modulate this signal?

We agree with the reviewer that the suggested questions are interesting. However, we feel that our paper, which contains a bulk of analyses throughout all possible task phases and is already very voluminous, would lose its focus. As acknowledged by the reviewer, we think these issues are beyond the scope of this study and we therefore would prefer not to include them.

REVIEWERS' COMMENTS

Reviewer #1 (Remarks to the Author):

The authors have meticulously addressed my comments in the revised manuscript and reviewer response. They have added new analyses which is shown in Figure 7 and extended table 1 and further, reworked their results and discussion in response to prior comments. I have no further concerns or comments.

Reviewer #2 (Remarks to the Author):

The authors have done a good job revising their manuscript. I am satisfied with all the additions and clarifications the authors made.